# Global COI meta-analysis reveals ocean-basin genetic structure in *Sphyrna lewini*

**Steph Smith** *, **Chelsea Black**

Department of Earth, Marine and Environmental Sciences, Institute of Marine Sciences, University of North Carolina, Morehead City, North Carolina, United States of America

* steph.smith@unc.edu

## Abstract

The scalloped hammerhead shark (*Sphyrna lewini*) is a circumglobally distributed apex predator clarified as Critically Endangered due to population declines exceeding 80%. Understanding population structure is essential for effective conservation management, yet comprehensive genetic surveys across the species' range remain limited. We conducted a meta-analysis of all publicly available cytochrome oxidase I (COI) sequences for the family Sphyrnidae from the Barcode of Life Database (BOLD), applying a hierarchical analytical approach that examined haplotype clustering patterns across all hammerhead species. This family-level analysis enabled detection of taxonomic inconsistencies invisible to single-species queries: 18 of 1,066 samples (1.7%) carried species labels inconsistent with their haplotype assignments, with *S. zygaena* comprising the majority of mislabeled specimens in *S. lewini* haplotypes. Among 595 confirmed *S. lewini* samples, we identified four haplotypes forming two distinct phylogenetic groups with 94.6–96.2% sequence identity across a 184 bp COI fragment. Geographic analysis of 65 *S. lewini* samples with precise coordinates revealed pronounced ocean basin-scale segregation. Haplotype Group A (h1 and h4) predominated in the Indian Ocean, Atlantic basins, and South Pacific, while Group B (h15 and h2) is centered in the western Pacific. Country-level metadata for all 595 samples supported this pattern, with Indonesia harboring both groups consistent with its position at the Indian-Pacific biogeographic boundary. The rare haplotype (h15; n = 5) restricted to the North Atlantic included depositor notes indicating "possible cryptic species". These findings reveal ocean basin-scale partitioning in *S. lewini* with implications for defining management units in this threatened species.

## Introduction

The family Sphyrnidae (hammerhead sharks) includes several species of conservation concern, with the scalloped hammerhead (*Sphyrna lewini*) listed as Critically Endangered by the International Union for Conservation of Nature (IUCN) due to severe population declines across all ocean basins [1]. This circumglobally distributed

**Data availability statement:** All sequences and metadata analyzed in this study are publicly available through BOLD using the taxonomy query 'Sphyrnidae'. Additionally, the complete analyzed dataset is provided as a Supporting information file. Code used to generate this analysis is available through BOLDGenotyper v1.0.0 (https://github.com/SymbioSeas/BOLDGenotyper), an open-source bioinformatics pipeline for automated COI genotyping and biogeographic analysis from BOLD data.

**Funding:** The author(s) received no specific funding for this work.

**Competing interests:** The authors have declared that no competing interests exist.

apex predator inhabits tropical to warm-temperature waters between 46°N and 36°S, ranging from surface waters to depths exceeding 500 m, with a maximum body length of approximately 4.2 m [2,3]. The scalloped hammerhead is distinguished from congeners by its broadly arched cephalofoil featuring a prominent central indentation flanked by smaller lateral notches [2]. Global hammerhead populations are estimated to have declined by over 80%, driven primarily by overfishing, bycatch mortality, habitat degradation, and climate change [4–6]. Despite their conservation significance, comprehensive understanding of population structure within *S. lewini* remains incomplete, limiting the development of effective, population-specific management strategies.

The family Sphyrnidae comprises two genera; *Sphyrna* (nine species), and *Eusphyra* (one species, the winghead shark *E. blochii*), which are united by their laterally expanded, dorsoventrally flattened heads (cephalofoils) [7]. This defining character exhibits significant variation across the family: the cephalofoil of the bonnethead shark (*S. tiburo*) measures approximately 18% of body length, while that of the winghead shark extends to 50% of body length [3,7]. The number of recognized hammerhead species has expanded from 7 to 10 species from 1967 to 2018, reflecting both taxonomic revision and the discovery of cryptic lineages [2,8–10]. Morphological similarity among the large-bodied hammerheads complicates field identification, particularly for juveniles in which diagnostic cephalofoil morphology may be less pronounced [2,11]. This identification challenge has important implications for both fisheries management and the accuracy of species records in public databases.

The family Sphyrnidae has emerged as a notable example of cryptic diversity among elasmobranchs. The Carolina hammerhead (*Sphyrna gilberti*), described from the western North Atlantic, is currently distinguished from *S. lewini* only in the number of precaudal vertebrae, yet is genetically divergent with a confirmed range from North Carolina to Florida [9]. These species hybridize in sympatry, with genomic analysis revealing first-generation hybrids and backcrosses along the U.S. Atlantic coast [12]. Similarly, the shovelbill shark (*Sphyrna alleni*) was recently described from the Caribbean following recognition of cryptic genetic divergence within the bonnethead complex [13]. This marks the second divergence within bonnethead sharks, following the species recognition of the Pacific bonnethead shark, *S. vespertina*. These discoveries highlight the potential for unrecognized diversity within broadly distributed hammerhead species and emphasize the importance of genetic approaches for accurate species delimitation and population assessment.

Marine species with apparently continuous distributions can harbor genetically distinct populations when dispersal is limited or when females exhibit philopatry to natal sites [14,15]. *Sphyrna lewini* demonstrates life history traits conducive to population subdivision: site fidelity to aggregation areas, utilization of coastal embayments as nursery habitat, and evidence of female philopatry limiting gene flow across oceanic barriers [16–18]. Previous phylogeographic investigations documented significant genetic structure across the species' range. Duncan et al. (2006) identified three mitochondrial lineages within *S. lewini* corresponding to ocean basins [19]. Subsequent mixed-marker analysis revealed that while maternal (mitochondrial) lineages

show strong structure, nuclear markers indicate male-mediated gene flow across oceanic basins, suggesting female philopatry with male dispersal [20]. More recent multi-marker approaches have confirmed these connectivity patterns in the Pacific and Indian Oceans [21,22].

The cytochrome c oxidase subunit I (COI) gene serves as the primary DNA barcode for metazoan species identification, and its effectiveness for distinguishing elasmobranch species and populations has been validated across multiple regional studies [23–25]. The Barcode of Life Data Systems (BOLD) now contains over 17 million COI sequences, providing unprecedented opportunities for large-scale comparative analyses across species' ranges [26,27]. However, public sequence databases may contain misidentified specimens that can confound population genetic analyses. This concern is particularly relevant for morphologically similar species like hammerheads, where field identification is challenging even for experienced researchers. Family-level comparative analysis examining clustering patterns across closely related taxa offers a quality control approach for detecting such errors that single-species queries cannot identify.

Given the conservation significance of *S. lewini* and the need for accurate genetic characterization of population structure, we conducted a meta-analysis of all publicly availably COI sequences for the family Sphyrnidae from BOLD. We employed a hierarchical analytical approach, examining haplotype clustering patterns across all hammerhead species to identify potential database misidentifications before characterizing genetic diversity within confirmed *S. lewini* samples. Our objectives were to: (1) evaluate the extent of taxonomic inconsistencies in BOLD Sphyrnidae records through family-level comparative analysis, (2) characterize the diversity and geographic distribution of *S. lewini* COI haplotypes, and (3) assess whether ocean basins represent significant barriers to maternal gene flow in this threatened species.

## Results

### Quality control and misidentification detection

Analysis of 1,066 Sphyrnidae COI sequences from BOLD identified 13 distinct haplotypes representing eight recognized hammerhead species within the family (Table 1). Family-level comparative analysis revealed taxonomic inconsistencies between haplotypes assignments and database species labels. We identified 18 samples (1.7%) where the reported species label disagreed with the haplotype majority species assignment (Table 2). Among these, 14 samples in *S. lewini*-majority haplotypes were labeled as other species, including *S. zygaena* (n = 6), *E. blochii* (n = 3), *S. mokarran* (n = 2), and *S. tiburo* (n = 3). Conversely, two *S. lewini*-labeled samples clustered with non-*S. lewini* haplotypes: one with *S. mokarran* and one with *S. tiburo*. Two additional cross-species inconsistencies were detected between *S. mokarran* and *S. zygaena*/*S. tiburo* haplotypes. Family-level comparative analysis revealed taxonomic inconsistencies between haplotype assignments and database species labels (S1 Fig in S1 File). These taxonomic inconsistencies would be undetectable through single-species database queries and demonstrate the value of family-level analysis for quality control in public sequence databases.

### Haplotype diversity in confirmed *S. lewini*

Among 595 confirmed *S. lewini* samples, four distinct haplotypes were identified that form two well-supported phylogenetic groups (Fig 1A). Haplotype Group A comprised h1 (n = 226; 38.0%) and h4 (n = 134; 22.5%), together representing 60.5% of confirmed *S. lewini* samples. Haplotype Group B comprised h15 (n = 5; 0.8%) and h2 (n = 230; 38.7%), representing the remaining 39.5% of *S. lewini* samples. Within-group uncorrected p-distances were low: 0.005 (99.5% sequence identity) between h1 and h4. Between-group p-distances ranged from 0.038 to 0.054 (94.6–96.2% identity), indicating substantial divergence between the two haplotype groups.

Two additional *S. lewini*-majority haplotypes (h11, n = 42; h13, n = 10) could not be aligned to the 184 bp COI fragment examined here due to sequence length limitations and were excluded from phylogenetic analysis. However, their presence suggests that additional cryptic diversity may exist within the *S. lewini* species complex.

**Table 1. Sphyrnidae haplotype species composition.**

| Haplotype ID | Total Samples | Species Count | BOLD Reported Species Composition | Primary Species (%) | Consensus Species |
|---|---|---|---|---|---|
| h1 | 239 | 3 | *Sphyrna lewini*: 226 (94.6%); *Sphyrna zygaena*: 6 (2.5%); *Eusphyra blochii*: 3 (1.3%) | 94.6% | *Sphyrna lewini* |
| h2 | 230 | 1 | *Sphyrna lewini*: 230 (100%) | 100% | *Sphyrna lewini* |
| h3 | 152 | 1 | *Sphyrna zygaena*: 151 (99.3%) | 99.3% | *Sphyrna zygaena* |
| h4 | 135 | 1 | *Sphyrna lewini*: 134 (99.3%) | 99.3% | *Sphyrna lewini* |
| h5 | 102 | 1 | *Sphyrna mokarran*: 101 (99.0%) | 99.0% | *Sphyrna mokarran* |
| h6 | 42 | 3 | *Sphyrna tiburo*: 40 (95.2%); *Sphyrna mokarran*: 1 (2.4%); *Sphyrna lewini*: 1 (2.4%) | 95.2% | *Sphyrna tiburo* |
| h11 | 42 | 3 | *Sphyrna lewini*: 25 (59.5%); *Sphyrna mokarran*: 10 (23.8%); *Sphyrna zygaena*: 7 (16.7%) | 59.5% | *Sphyrna lewini* |
| h8 | 34 | 1 | *Sphyrna tiburo*: 34 (100%) | 100% | *Sphyrna tiburo* |
| h9 | 23 | 1 | *Sphyrna tudes*: 23 (100%) | 100% | *Sphyrna tudes* |
| h10 | 19 | 1 | *Eusphyra blochii*: 19 (100%) | 100% | *Eusphyra blochii* |
| h13 | 10 | 2 | *Sphyrna lewini*: 6 (60.0%); *Sphyrna zygaena*: 4 (40.0%) | 60% | *Sphyrna lewini* |
| h15 | 5 | 1 | *Sphyrna lewini*: 5 (100%) | 100% | *Sphyrna lewini* |
| h17 | 4 | 1 | *Sphyrna corona*: 4 (100%) | 100% | *Sphyrna corona* |

*Haplotypes h11 and h13 could not be aligned to the 184 bp COI fragment and were excluded from downstream phylogenetic analysis.

**Table 2. Potential misidentifications in BOLD Sphyrnidae records.**

| BOLD processid (sample ID) | BOLD Reported Species | Haplotype ID | Haplotype Majority Species | Haplotype Majority Species (%) |
|---|---|---|---|---|
| ANGBF47378−19 | *Eusphyra blochii* | h1 | *Sphyrna lewini* | 94.6% |
| ANGBF47379−19 | *Eusphyra blochii* | | | |
| GBMNA14268−19 | *Eusphyra blochii* | | | |
| GBMNA14259−19 | *Sphyrna mokarran* | | | |
| GBMNA17866−19 | *Sphyrna mokarran* | | | |
| GBMNA14257−19 | *Sphyrna tiburo* | | | |
| GBMTG5888−16 | *Sphyrna tiburo* | | | |
| ANGBF16076−19 | *Sphyrna zygaena* | | | |
| ANGBF16077−19 | *Sphyrna zygaena* | | | |
| GBGC15811−19 | *Sphyrna zygaena* | | | |
| GBGC15812−19 | *Sphyrna zygaena* | | | |
| GBMNA14258−19 | *Sphyrna zygaena* | | | |
| GBMTG4448−16 | *Sphyrna zygaena* | | | |
| ANGBF12570−15 | *Sphyrna mokarran* | h3 | *Sphyrna zygaena* | 99.3% |
| GBMNF42720−22 | *Sphyrna tiburo* | h4 | *Sphyrna lewini* | 99.3% |
| GBMNF42715−22 | *Sphyrna lewini* | h5 | *Sphyrna mokarran* | 99.0% |
| MXIV374−10 | *Sphyrna lewini* | h6 | *Sphyrna tiburo* | 95.2% |
| GBMNF42716−22 | *Sphyrna mokarran* | | | |

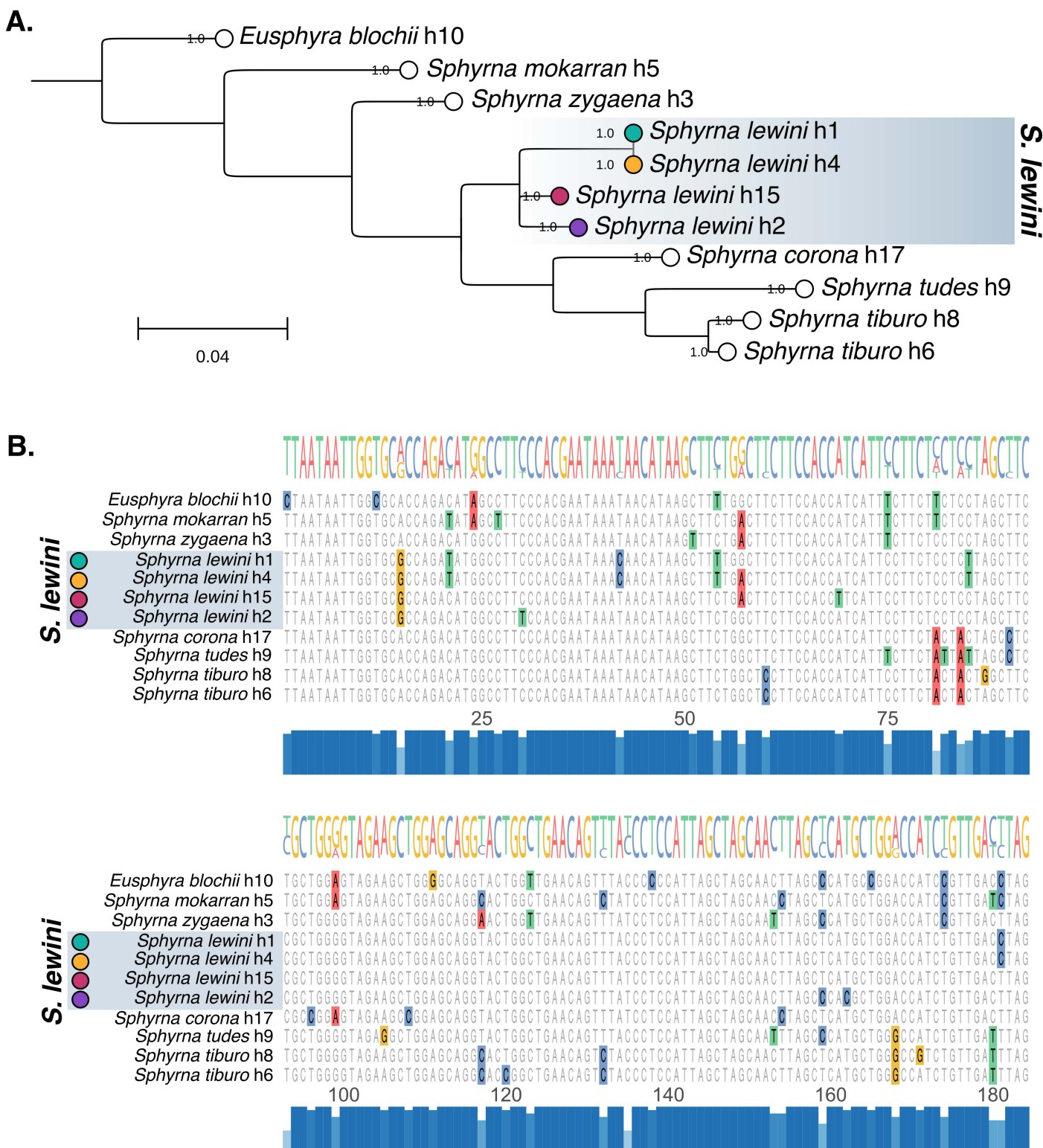

**Fig 1. Phylogenetic relationships and sequence divergence among Sphyrnidae COI haplotypes. (A)** Maximum likelihood phylogeny of Sphyrnidae COI haplotype consensus sequences constructed using PhyML with GTR model and 1,000 bootstrap replicates. Bootstrap support values are indicated at nodes. The four *S. lewini* haplotypes (highlighted) form two distinct groups: Haplotype Group A (h1, teal; h4, yellow), and Haplotype Group

B (h15, pink; h2, purple). Scale bar indicates substitutions per site. **(B)** Multiple sequence alignment of the 184 bp COI fragment showing all Sphyrnidae haplotypes. Polymorphic sites are highlighted; *S. lewini* haplotypes are indicated by colored circles corresponding to panel **A.** Conservation histogram shown below alignment.

## Geographic distribution patterns

Geographic analysis was limited by metadata availability; only 65 confirmed *S. lewini* samples (10.9%) had precise coordinates that mapped to defined ocean basins (Fig 2). Despite this limitation, strong ocean basin-scale segregation between haplotype groups was evident (Fig 2B-C).

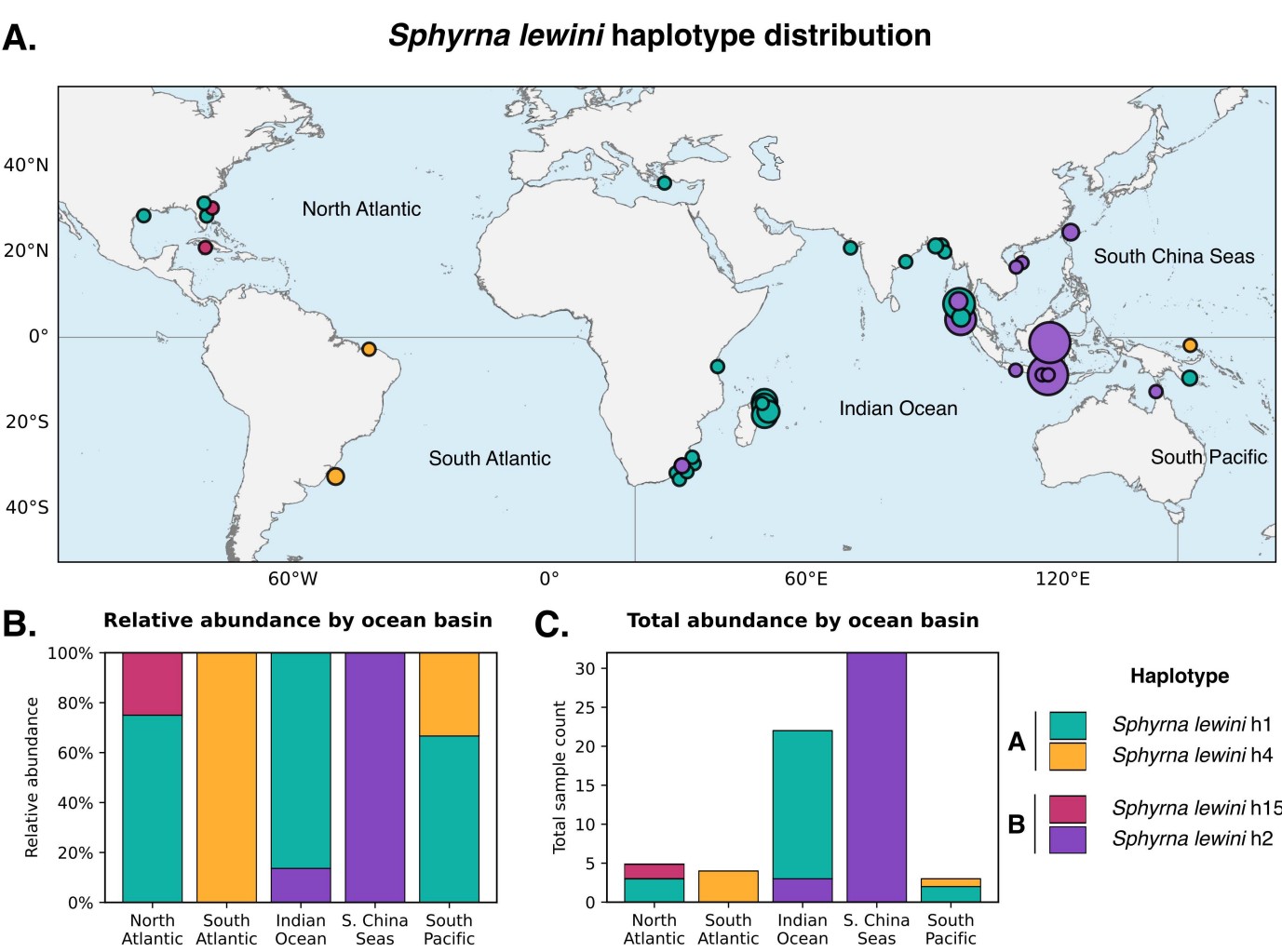

**Fig 2. Global distribution patterns of *Sphyrna lewini* COI haplotypes. (A)** Geographic coordinates of 65 confirmed *S. lewini* samples with precise ocean basin assignments, color-coded by haplotype (h1, teal; h2, purple; h4, yellow; h15, pink). Circle size corresponds to sample count at each coordinate. Ocean basin boundaries indicated. **(B)** Relative abundance of *S. lewini* haplotypes within each ocean basin, showing clear dominance of Haplotype Group A (h1 and h4) in the Indian Ocean and Atlantic basins, and Haplotype B (h2) in the South China Seas. **(C)** Absolute sample counts by ocean basin and haplotype.

Haplotype Group A (h1 and h4) predominated in the Indian Ocean and Atlantic basins. Of 24 h1 samples with ocean basin assignments, 19 (70.2%) originated from the Indian Ocean, with additional representation in the North Atlantic (n = 3) and South Pacific (n = 2). All five h4 samples with coordinates were from Atlantic basins: South Atlantic (n = 4) and South Pacific (n = 1). Haplotype h15 was constrained to the North Atlantic, while haplotype h2 dominated the western Pacific region. Of 35 h2 samples with coordinates, 32 (91.4%) originated from the South China Seas, with minor representation in the Indian Ocean (n = 3). The single h15 sample with precise coordinates was from the North Atlantic.

Country-level metadata, available for all 595 confirmed *S. lewini* samples, broadly supported these ocean basin patterns despite the inherent imprecision of country-level geographic data for multi-basin nations. Haplotype Group A samples (n = 360) originated primarily from Madagascar (n = 66), Brazil (n = 31), Indonesia (n = 27), and India (n = 14). Haplotype h2 samples (n = 235) were concentrated in Indonesia (n = 122), Taiwan (n = 25), Australia (n = 18), and Papua New Guinea (n = 15). Notably, Indonesia harbored both haplotype groups, with h2 predominating (n = 122) over h1 (n = 27), consistent with this archipelago's position at the biogeographic boundary between the Indian Ocean and western Pacific provinces.

The rare h15 haplotype (n = 5) was restricted to the western Atlantic, with samples originating from the United States (n = 2) and unspecified Atlantic Ocean locations (n = 2). Notably, three of five h15 samples included depositor notes indicating "possible cryptic species", suggesting that original collectors recognized morphological distinctiveness in these specimens (Table 3).

## Phylogenetic relationships

Maximum likelihood phylogenetic analysis of the COI fragment confirmed that *S. lewini* haplotypes form a monophyletic group within Sphyrnidae, with all four haplotypes clustering together with strong bootstrap support (Fig 1A). Within *S. lewini*, the haplotypes resolve into two well-supported mitochondrial clades. Haplotype Group A comprised h1 and h4, while Haplotype Group B comprised h2 and the rare h15 lineage. These two groups were reciprocally monophyletic and positioned as sister to *S. corona* within the broader Sphyrnidae phylogeny.

Pairwise sequence analysis across the COI fragment revealed 7 polymorphic sites distinguishing the four *S. lewini* haplotypes (Fig 1B). Divergence between the two *S. lewini* mitochondrial groups was substantial; the maximum between-group p-distance (0.054, between h4 and h15) equaled the interspecific distance separating *S. lewini* (h15) from *S. corona*. For context, interspecific p-distances within Sphyrnidae ranged from 0.054 to 0.120 across the 184 bp COI fragment. Uncorrected pairwise p-distances among all haplotypes are shown in S2 Fig in S1 File.

## Discussion

### Database quality and methodological implications

Our family-level comparative analysis revealed that 1.7% of Sphyrnidae samples in BOLD carried species labels inconsistent with their haplotype assignments. This finding has important methodological implications for database-driven population genetic studies. Single-species queries are the standard approach for many meta-analyses, but they cannot detect such errors because they lack the comparative framework to identify samples whose sequences cluster with congeners

**Table 3. BOLD depositor notes suggesting cryptic *S. lewini* diversity.**

| BOLD processid (sample ID) | Haplotype ID | Haplotype Species | BOLD Depositor Note |
|---|---|---|---|
| EWHHS105−06 | h15 | *Sphyrna lewini* | Tube #104, possible cryptic species |
| EWHHS106−06 | h15 | *Sphyrna lewini* | Tube #105, possible cryptic species |
| EWHHS107−06 | h15 | *Sphyrna lewini* | Tube #106, possible cryptic species |

rather than conspecifics. The morphological similarity among hammerhead species likely contributes to field identification errors that propagate into sequence databases [2]. Misidentification between *S. lewini* and *S. mokarran* has been recorded, particularly in juveniles when morphological differences in cephalofoil shape may not be as apparent. While 1.7% represents a modest error rate, even small proportions of misidentified samples can confound population structure analyses, particularly when sample sizes from specific regions are limited [11]. Our approach demonstrates that family-level analysis provides an effective quality control mechanism for meta-analyses using BOLD and similar repositories.

## Ocean basin-scale genetic structure

The marked geographic segregation between *S. lewini* haplotype groups corroborates and extends previous phylogeographic findings. Duncan et al. (2006) identified three mitochondrial lineages within *S. lewini* using control region sequences, with strong population subdivision corresponding to ocean basins and evidence of late Pleistocene radiations from an Indo-Pacific center of origin [19]. More recently, two distinct *S. lewini* clades were identified using COI barcoding of specimens collected from Malaysian waters [25]. Our COI-based analysis reveals a similar pattern: Haplotype Group A (h1 and h4) predominates in the Indian Ocean and Atlantic basins, while Haplotype Group B (h15 and h2) dominates the South China Seas and western Pacific. The between-group sequence divergence of 3.8–5.4% indicates substantial evolutionary divergence, with the maximum p-distance (0.054) equaling the interspecific distance between *S. lewini* and *S. corona* across the COI fragment examined here.

These patterns align with broader biogeographic principles in which ocean basins act as effective barriers to gene flow even for highly mobile marine species [19,28]. Daly-Engel et al. (2012) demonstrated that while mitochondrial markers show strong structure in *S. lewini*, nuclear markers indicate male-mediated gene flow across oceanic expanses [20]. Our findings are thus consistent with female philopatry limiting maternal lineage dispersal across ocean basins, with males likely facilitating genetic connectivity that maintains species cohesion despite mitochondrial divergence. The co-occurrence of both haplotype groups in Indonesia, positioned at the boundary between Indian Ocean and western Pacific biogeographic provinces, further supports this interpretation and identifies the region as a potential secondary contact zone.

## The rare North Atlantic haplotype

The rare h15 haplotype, represented by only five samples and restricted to the western Atlantic, merits particular attention. Three of five h15 samples included depositor notes indicating "possible cryptic species", suggesting that collectors recognized morphological distinctiveness in these specimens at the time of deposition. This observation is intriguing given the Atlantic-Caribbean region's propensity for harboring cryptic hammerhead diversity: *S. gilberti*, described from the western Atlantic and morphologically indistinguishable from *S. lewini* except in vertebral counts, was confirmed as a distinct species through genetic analysis [9], and *S. alleni* was similarly recognized within the bonnethead complex from Caribbean waters [13].

Alignment of h15 to a reference *S. gilberti* COI sequence extracted from the genome assembly reported by Barker et al. (2019) [NCBI Accession KY315827.1] revealed 100% nucleotide identity across the COI fragment examined here [12]. This sequence identity, combined with the geographic restriction of h15 to the western North Atlantic (consistent with the documented *S. gilberti* range from North Carolina to Florida), raises the possibility that h15 represents *S. gilberti* maternal lineages rather than *S. lewini*. However, several caveats preclude definitive taxonomic assignment. COI is a maternally inherited marker that cannot distinguish pure species from hybrids carrying *S. gilberti* mitochondria; given documented hybridization between these species [12], some h15 individuals could represent backcrossed hybrids. Nevertheless, this preliminary evidence suggests that h15 warrants targeted investigation with nuclear markers and morphological examination to resolve its taxonomic status.

Sphyrnidae diverged from Carcharhiniformes only 10–20 million years ago, making hammerheads among the youngest extant shark families [7,29]. Given this recent radiation and the ongoing discovery of cryptic species within the family, additional genetically distinct lineages likely remain unrecognized.

## Conservation implications

The genetic partitioning documented here has direct relevance for *S. lewini* conservation. Current management approaches treating the species as a single global unit may overlook population-level differences affecting vulnerability and recovery capacity [30]. The distinct haplotype groups identified in this study may warrant recognition as separate management units, particularly given the species' Critically Endangered status and estimated population declines exceeding 80% globally [1,4–6]. Haplotype Group A, distributed across the Indian Ocean and Atlantic basins, and Haplotype Group B, concentrated in the western Pacific and South China Seas, appear to represent demographically and evolutionarily semi-independent units that may be connected primarily through male-mediated gene flow. Indonesia's harboring of both *S. lewini* haplotype groups highlights this archipelago's importance as both a contact zone and conservation priority area where management actions could affect multiple evolutionary lineages. Given the strong mitochondrial divergence and basin-restricted haplotypes, these groups may represent evolutionarily significant subunits (ESUs) under several conservation frameworks.

## Limitations and future directions

Only 10.9% of confirmed *S. lewini* samples had precise coordinates suitable for ocean basin assignment, limiting geographic inference. Our analyses focused on a conservative 184 bp COI fragment to retain the maximum number of samples with precise geographic metadata, which was already limited in the source dataset. This shortened region is less variable than the full-length COI barcode typically used in population genetic studies; however, its use did not obscure major phylogeographic structure. In exploratory analyses using a longer 591 bp COI fragment which excluded more samples lacking complete sequence coverage, we recovered an even greater number of unique haplotypes that nonetheless clustered into the same divergent *S. lewini* clades identified here. These consistent patterns indicate that ocean-basin partitioning we report is robust to marker length and not an artifact of the conservative alignment region.

While country-level metadata for all 595 samples broadly supported the ocean basin patterns, this coarser geographic resolution is imprecise for multi-basin nations and was therefore treated as supporting rather than primary evidence. This 184 bp COI fragment, while sufficient to distinguish species and major haplotype groups within Sphyrnidae, provides limited phylogenetic resolution compared to full-length barcodes or multi-locus datasets. COI represents a single maternally inherited marker; integration with nuclear loci is essential for comprehensive population assessment and to distinguish true population structure [21]. Additionally, two *S. lewini* haplotypes (h11, h13) could not be aligned to the COI fragment examined here, and their taxonomic status remains unresolved.

A further limitation stems from the taxonomic information available in public repositories. Several cryptic hammerhead species described in recent years – notably *Sphyrna gilberti* and *Sphyrna alleni* – are not represented as distinct taxonomic categories within BOLD. As a result, the present analysis can identify unique and phylogenetically coherent haplotypes that likely correspond to divergent lineages, but it cannot assign species names to haplotypes belonging to taxa that were not recognized at the time of sequence deposition. Thus, sequences belonging to *S. gilberti* would be correctly placed phylogenetically but would appear under *S. lewini* haplotypes in this dataset due to historical labeling in BOLD. These constraints highlight the need for updated taxonomic annotation in public databases and for future studies integrating morphometric, nuclear genomic, and newly recognized species boundaries when interpreting mitochondrial haplotype structure.

Future studies should prioritize expanded geographic sampling with precise GPS coordinates, multi-locus genetic analysis integrating nuclear markers, and morphometric comparisons between haplotype groups. The depositor notes associated with the h15 haplotype suggest that field biologists have recognized phenotypic variation within *S. lewini* that may correspond to the genetic structure documented here. Systematic morphological analysis across the species' range, informed by haplotype assignments, could reveal diagnostic characters useful for field identification and conservation monitoring.

 

## Conclusions

This meta-analysis of publicly available COI sequences reveals ocean basin-scale genetic partitioning within *S. lewini*, with two dominant haplotype groups exhibiting 94.6–96.2% identity and largely non-overlapping geographic distributions. Family-level comparative analysis enabled detection of taxonomic inconsistencies in database records while simultaneously characterizing intraspecific diversity. These findings support recognition of Indian Ocean-Atlantic and western Pacific management units for this Critically Endangered species and demonstrate the utility of hierarchical analytical approaches for database-driven phylogeographic research.

## Methods

### Data acquisition

All publicly available COI sequences and associated metadata for the family Sphyrnidae were downloaded from the Barcode of Life Database (BOLD; www.boldsystems.org) on November 4th, 2025 using the query [Sphyrnidae] [27]. Family-level analysis was employed rather than species-specific queries to enable detection of potential taxonomic misidentifications through comparative analysis across related species. The initial download yielded 1,108 records; following removal of duplicate process IDs and samples lacking sequence data, 1,066 samples remained for analysis.

### Sequence processing and haplotype identification

Sequence quality control, alignment, and haplotype identification were performed using custom python scripts. Sequences were aligned using MAFFT v7 [31] with automatic algorithm selection. A core COI region of 184 bp, representing positions with ≥80% sequence coverage across all samples, was extracted for downstream analysis to accommodate the variable sequence lengths present in BOLD (ranging from approximately 400–700 bp in the raw dataset). Haplotypes were defined as exact sequence variants (ESVs) within this fragment; sequences sharing identical COI fragment sequences were assigned to the same haplotype regardless of variation in flanking regions. This 184 bp fragment was selected to maximize sample retention and preserve geographic metadata coverage, as longer COI fragments would have excluded a substantial fraction of samples with usable coordinates.

### Taxonomic assignment and misidentification detection

Taxonomic analysis aggregated species labels within each haplotype to determine the majority species assignment. Haplotypes were assigned to the species comprising >50% of member samples. Samples were classified as confirmed *S. lewini* if they (1) belonged to haplotypes where *S. lewini* was the majority species and (2) carried concordant *S. lewini* labels in BOLD. Samples where the database species label disagreed with the haplotype majority species were flagged as potential misidentifications. Depositor notes fields were examined for keywords indicating taxonomic uncertainty (e.g., "cryptic", "uncertain").

### Geographic analysis

Geographic coordinates were parsed from BOLD metadata and validated for quality. Samples with missing coordinates, coordinates corresponding to country centroids (indicating imprecise location data), or coordinates falling on land were excluded from ocean basin analysis. Remaining samples were assigned to ocean basins using spatial intersection with the Global Oceans and Seas (GOaS) v1 reference shapefile [32]. Country-level metadata was summarized separately for all confirmed *S. lewini* samples to provide supporting phylogeographic context, with the caveat that country-level data is imprecise for nations spanning multiple ocean basins.

### Phylogenetic analysis

Maximum likelihood phylogenetic analysis was conducted on haplotype consensus sequences using PhyML v3.0 [33] with the GTR nucleotide substitution model. Branch support was estimated from 1,000 bootstrap replicates. Uncorrected

pairwise p-distances between haplotypes were calculated from the 184 bp COI alignment to quantify within-species and between-species divergence. Haplotypes that could not be aligned to this fragment due to incomplete sequence coverage were excluded from phylogenetic analysis but retained in taxonomic summaries.

## Supporting information

**S1 File. Supporting figures S1 and S2.**
(DOCX)

**S2 File. Sphyrnidae_annotated_metadata.csv.** This file contains the complete annotated BOLD dataset used for analysis.
(CSV)

## Acknowledgments

We thank the researchers and institutions who generated and deposited the publicly available COI sequences analyzed in this study. Their commitment to open data made this work possible.

## Author contributions

**Conceptualization:** Steph Smith, Chelsea Black.

**Data curation:** Steph Smith.

**Formal analysis:** Steph Smith.

**Investigation:** Steph Smith, Chelsea Black.

**Methodology:** Steph Smith.

**Visualization:** Steph Smith.

**Writing – original draft:** Steph Smith.

**Writing – review & editing:** Chelsea Black.

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
