## [Decision Letter · Decision Letter 0]

3 Feb 2026

Dear Dr. Smith,

Thank you for submitting your manuscript to PLOS ONE. After careful consideration, we feel that it has merit but does not fully meet PLOS ONE’s publication criteria as it currently stands. Therefore, we invite you to submit a revised version of the manuscript that addresses the points raised during the review process.

We look forward to receiving your revised manuscript.

Kind regards,

Karla Diamantina de Araújo Soares

Academic Editor

PLOS One

Journal Requirements:

3. We note that Figure 2 in your submission contain [map/satellite] images which may be copyrighted. All PLOS content is published under the Creative Commons Attribution License (CC BY 4.0), which means that the manuscript, images, and Supporting Information files will be freely available online, and any third party is permitted to access, download, copy, distribute, and use these materials in any way, even commercially, with proper attribution. For these reasons, we cannot publish previously copyrighted maps or satellite images created using proprietary data, such as Google software (Google Maps, Street View, and Earth). For more information, see our copyright guidelines: http://journals.plos.org/plosone/s/licenses-and-copyright .

(1) You may seek permission from the original copyright holder of Figure 2 to publish the content specifically under the CC BY 4.0 license.

4. Please include captions for your Supporting Information files at the end of your manuscript, and update any in-text citations to match accordingly. Please see our Supporting Information guidelines for more information: http://journals.plos.org/plosone/s/supporting-information .

Reviewers' comments:

Reviewer's Responses to Questions

**Comments to the Author**

1. Is the manuscript technically sound, and do the data support the conclusions?

Reviewer #1: Yes

Reviewer #2: Yes

2. Has the statistical analysis been performed appropriately and rigorously?

Reviewer #1: Yes

Reviewer #2: N/A

3. Have the authors made all data underlying the findings in their manuscript fully available?

Reviewer #1: Yes

Reviewer #2: Yes

4. Is the manuscript presented in an intelligible fashion and written in standard English?

Reviewer #1: Yes

Reviewer #2: Yes

Reviewer #1: 1. I suggest standardizing the taxonomic presentation by following the conventional order from family to species throughout the manuscript. E.g. Sphyrnadae, Sphyrna lewini

2. The keywords ‘COI barcoding’ and ‘DNA barcoding’ appear to be redundant, since COI is the standard marker used for DNA barcoding. I suggest keeping only one of them to avoid redundancy.

3. Lines 42- 48 and lines 49 - 51 - I suggest reorganizing the first two paragraphs to follow a taxonomic hierarchy, introducing the family Sphyrnidae before focusing on the target species, Sphyrna lewini.

4. I suggest reorganizing the paragraphs to avoid switching between Sphyrna lewini and other species, which disrupts the narrative flow. Presenting all information on S. lewini together would improve clarity. For example, information on S. lewini appears in lines 42–48 and again in lines 62–71. Reorganizing these paragraphs to keep species-specific information together would improve readability.

5. Although the objectives are clearly defined (taxonomic inconsistencies, haplotype diversity and distribution, and ocean-basin structure), the presentation of information across the paragraphs appears fragmented. Reorganizing the paragraphs to follow the same sequence as the stated objectives would help link the narrative more clearly to each objective.

6. Lines 118 and 122- In Figure 1, please revise the legend for clarity and formatting. The term “core COI” is ambiguous and not standard. Additionally, family-level taxonomic names such as Sphyrnidae should not be italicized, as italics are reserved for genus and species names. Please note that this formatting issue occurs not only in the figure legends but throughout the manuscript, including Figure S1.

7. Although the sequences correspond to the COI gene, the analyzed region comprises only 184 bp, which is considerably shorter than the standard COI barcode region (~650 bp). Therefore, referring to this marker as “COI region” may be misleading. We suggest using the term “COI fragment” (or “partial COI fragment”) to more accurately reflect the length of the analyzed sequences.

8. The term “core COI” is repeatedly used throughout the manuscript. While its use may be justified in the Methods section to describe the nature of the BOLD metadata and the short sequence length analyzed, it is unnecessary in the Results and Discussion. I suggest using simply COI, or “the 184 bp COI fragment analyzed here” when sequence length is relevant, to improve clarity and consistency.

Reviewer #2: This manuscript presents a global meta-analysis of publicly available COI sequences to investigate ocean-basin–scale genetic structure in Sphyrna lewini. The study addresses a timely and relevant topic, given the species’ current Critically Endangered status, and demonstrates the utility of family-level comparative analyses for identifying identification inconsistencies in public databases. The overall approach is sound, and the results have clear implications for conservation and management. The following comments are intended to improve clarity, taxonomic context, and the presentation of the manuscript.

**Do you want your identity to be public for this peer review?** For information about this choice, including consent withdrawal, please see our Privacy Policy

Reviewer #1: No

Reviewer #2: No

---

## [Author Response · Author response to Decision Letter 1]

10 Feb 2026

PONE-D-25-65513

Global COI meta-analysis reveals ocean-basin genetic structure in Sphyrna lewini

PLOS One

EDITOR NOTES

The PLOS ONE style templates can be found at

The manuscript style, citation style, and file naming conventions have been updated to reflect the templates linked above.

Code used to generate this analysis is available from the corresponding author SNS upon request, and this has been noted in the Data Availability statement. Note that we are preparing to submit a complementary manuscript that provides the complete workflow for this analysis as a comprehensive Python package. This follow-up manuscript will demonstrate the use of the ‘BOLDGenotyper’ package on several marine, freshwater, and terrestrial datasets, and will reference this manuscript as a case study. At that time, all code will be readily available on GitHub associated with SNS (@SymbioSeas).

3. We note that Figure 2 in your submission contain [map/satellite] images which may be copyrighted. All PLOS content is published under the Creative Commons Attribution License (CC BY 4.0), which means that the manuscript, images, and Supporting Information files will be freely available online, and any third party is permitted to access, download, copy, distribute, and use these materials in any way, even commercially, with proper attribution. For these reasons, we cannot publish previously copyrighted maps or satellite images created using proprietary data, such as Google software (Google Maps, Street View, and Earth). For more information, see our copyright guidelines: http://journals.plos.org/plosone/s/licenses-and-copyright.

We confirm that the Global Oceans and Seas (GOaS) v1 shapefile used to generate Figure 2 is published under a Creative Commons Attribution 4.0 International license (CC BY 4.0), which permits sharing and adaptation for any purpose, including commercial use, with appropriate attribution. This shapefile is cited in our manuscript as Reference 31 (Flanders Marine Institute, 2021), and the Creative Commons license is documented in the source repository [https://www.vliz.be/en/imis?dasid=7842&doiid=613]. No additional permissions are required for publication or reproduction.

We have added a Supporting Information section at the end of the manuscript following References, with complete captions for both supplementary figures. In-text citations have been updated to reference “S1 Fig” and “S2 Fig” per PLOS ONE style requirements.

REVIEWER 1

1. I suggest standardizing the taxonomic presentation by following the conventional order from family to species throughout the manuscript. E.g. Sphyrnadae, Sphyrna lewini.

We have updated taxonomic presentation throughout the Introduction as suggested. This is expanded on in our response to comment #3.

2. The keywords ‘COI barcoding’ and ‘DNA barcoding’ appear to be redundant, since COI is the standard marker used for DNA barcoding. I suggest keeping only one of them to avoid redundancy.

We have replaced the keyword “DNA barcoding” with “haplotyping”.

3. Lines 42- 48 and lines 49 - 51 - I suggest reorganizing the first two paragraphs to follow a taxonomic hierarchy, introducing the family Sphyrnidae before focusing on the target species, Sphyrna lewini.

We have revised the Introduction to establish taxonomic context earlier. The opening sentence now introduces the family Sphyrnidae before narrowing focus to our target species, S. lewini. The second paragraph now expands on family-level taxonomy, including the two recognized genera, morphological diversity across species, and identification challenges. We believe this reorganized Introduction balances taxonomic hierarchy with clarity of our narrative and addresses all reviewer comments.

4. I suggest reorganizing the paragraphs to avoid switching between Sphyrna lewini and other species, which disrupts the narrative flow. Presenting all information on S. lewini together would improve clarity. For example, information on S. lewini appears in lines 42–48 and again in lines 62–71. Reorganizing these paragraphs to keep species-specific information together would improve readability.

We appreciate this feedback regarding narrative flow. We have reorganized the Introduction so that information on our focal species (S. lewini) is now consolidated in two sequential paragraphs: first establishing its conservation status and significance, then describing its life history traits relevant to population structure (site fidelity, philopatry, nursery habitat use). The discussion of cryptic diversity in other hammerhead species (S. gilberti, S. alleni) now follows as context for potential unrecognized diversity.

5. Although the objectives are clearly defined (taxonomic inconsistencies, haplotype diversity and distribution, and ocean-basin structure), the presentation of information across the paragraphs appears fragmented. Reorganizing the paragraphs to follow the same sequence as the stated objectives would help link the narrative more clearly to each objective.

We thank the reviewer for this helpful structural suggestion. We have reorganized the Introduction to follow the logical sequence of our stated objectives. The revised introduction now progresses from: (1) general species introduction and conservation significance, (2) taxonomic background on the family Sphyrnidae and challenges in species identification, (3) cryptic diversity discoveries within hammerheads, (4) population structure and maternal gene flow limitations in S. lewini, and (5) the utility of COI barcoding and meta-analysis approaches. Line-by-line changes are documented in the revised manuscript.

6. Lines 118 and 122- In Figure 1, please revise the legend for clarity and formatting. The term “core COI” is ambiguous and not standard. Additionally, family-level taxonomic names such as Sphyrnidae should not be italicized, as italics are reserved for genus and species names. Please note that this formatting issue occurs not only in the figure legends but throughout the manuscript, including Figure S1.

Italics have been removed from Sphyrnidae throughout the manuscript text and figure legends (including supplemental). Additionally, figure annotations were updated to ensure proper italicization of genus and species-level names in figures.

7. Although the sequences correspond to the COI gene, the analyzed region comprises only 184 bp, which is considerably shorter than the standard COI barcode region (~650 bp). Therefore, referring to this marker as “COI region” may be misleading. We suggest using the term “COI fragment” (or “partial COI fragment”) to more accurately reflect the length of the analyzed sequences.

See response below, this terminology has been clarified throughout the manuscript.

8. The term “core COI” is repeatedly used throughout the manuscript. While its use may be justified in the Methods section to describe the nature of the BOLD metadata and the short sequence length analyzed, it is unnecessary in the Results and Discussion. I suggest using simply COI, or “the 184 bp COI fragment analyzed here” when sequence length is relevant, to improve clarity and consistency.

The terminology "core COI" was intended to describe the aligned fragment that was "core" to the sequences analyzed here. However, we can appreciate that this is not standard terminology and have replaced "core COI" with "COI fragment", "COI region", or "COI fragment alignment" throughout the manuscript text and figure legends.

REVIEWER 2

This manuscript presents a global meta-analysis of publicly available COI sequences to investigate ocean-basin–scale genetic structure in Sphyrna lewini. The study addresses a timely and relevant topic, given the species’ current Critically Endangered status, and demonstrates the utility of family-level comparative analyses for identifying identification inconsistencies in public databases. The overall approach is sound, and the results have clear implications for conservation and management. The following comments are intended to improve clarity, taxonomic context, and the presentation of the manuscript.

Introduction

The Introduction provides limited taxonomic information on the family Sphyrnidae, the genera (Eusphyra and Sphyrna), and the focal species. In addition, it does not sufficiently address the taxonomic issues currently recognized within the genus Sphyrna.

We agree that additional taxonomic context strengthens the manuscript. We have added a new paragraph providing comprehensive background on the family Sphyrnidae, including its two recognized genera (Sphyrna and Eusphyra), the morphological diversity in cephalofoil structure across species, and the historical fluctuation in recognized species number from 7 to 10 species between 1967 and 2018. We have also expanded the description of our focal species S. lewini, including its distinguishing morphological characteristics (scalloped cephalofoil with central and lateral indentations), distribution (circumglobal, 46°N to 36°S), habitat preferences, and maximum body size. These additions are supported by citations to Lim et al. (2010), Compagno (1988), and Gallagher and Klimley (2018).

Line 87: In objective (1), replace “hammerhead” with “Sphyrnidae”.

Suggested revision: Evaluate the extent of taxonomic inconsistencies in BOLD Sphyrnidae records through comparative analysis.

This sentence has been revised as suggested.

Results

The tables presented in the text should be properly formatted.

Table formatting has been improved for clarity throughout.

Discussion

Line 179: “The morphological similarity among hammerhead species, particularly between S. lewini, S. zygaena, and S. mokarran, likely contributes to field identification errors that propagate into sequence databases.”

I suggest removing “particularly between S. lewini, S. zygaena, and S. mokarran”, as these species do not exhibit close morphological similarity.

Suggested revision: The morphological similarity among hammerhead species likely contributes to field identification errors that propagate into sequence databases.

This sentence has been revised as suggested.

---

## [Editor Report · Decision Letter 1]

26 Feb 2026

Global COI meta-analysis reveals ocean-basin genetic structure in *Sphyrna lewini*

PONE-D-25-65513R1

Dear Dr. Steph Smith,

We’re pleased to inform you that your manuscript has been judged scientifically suitable for publication and will be formally accepted for publication once it meets all outstanding technical requirements.

Kind regards,

Karla Diamantina de Araújo Soares

Academic Editor

PLOS One

---

## [Editor Report · Acceptance letter]

PONE-D-25-65513R1

PLOS One

Dear Dr. Smith,

I'm pleased to inform you that your manuscript has been deemed suitable for publication in PLOS One. Congratulations! Your manuscript is now being handed over to our production team.

Kind regards,

on behalf of

Dr. Karla Diamantina de Araújo Soares

Academic Editor

PLOS One